# Exploratory Full-Field Mechanical Analysis across the Osteochondral Tissue—Biomaterial Interface in an Ovine Model

**DOI:** 10.3390/ma13183911

**Published:** 2020-09-04

**Authors:** Jeffrey N. Clark, Agathe Heyraud, Saman Tavana, Talal Al-Jabri, Francesca Tallia, Brett Clark, Gordon W. Blunn, Justin P. Cobb, Ulrich Hansen, Julian R. Jones, Jonathan R. T. Jeffers

**Affiliations:** 1Department of Mechanical Engineering, Imperial College London, South Kensington Campus, London SW7 2AZ, UK; j.clark14@imperial.ac.uk (J.N.C.); s.tavana17@imperial.ac.uk (S.T.); u.hansen@imperial.ac.uk (U.H.); 2Department of Materials, Imperial College London, South Kensington Campus, London SW7 2AZ, UK; agathe.heyraud13@imperial.ac.uk (A.H.); f.tallia@imperial.ac.uk (F.T.); julian.r.jones@imperial.ac.uk (J.R.J.); 3Department of Surgery and Cancer, Imperial College London, London SW7 2AZ, UK; talal.al-jabri10@imperial.ac.uk (T.A.-J.); j.cobb@imperial.ac.uk (J.P.C.); 4Imaging and Analysis Centre, Natural History Museum London, London SW7 5BD, UK; brett.clark@nhm.ac.uk; 5School of Pharmacy and Biomedical Science, University of Portsmouth, Portsmouth PO1 2DT, UK; gordon.blunn@port.ac.uk

**Keywords:** biomaterials, cartilage regeneration, digital volume correlation, tissue-biomaterial interface, tissue regeneration, micro-CT, in situ mechanics, X-ray computed tomography, phase-contrast imaging

## Abstract

Osteochondral injuries are increasingly prevalent, yet success in articular cartilage regeneration remains elusive, necessitating the development of new surgical interventions and novel medical devices. As part of device development, animal models are an important milestone in illustrating functionality of novel implants. Inspection of the tissue-biomaterial system is vital to understand and predict load-sharing capacity, fixation mechanics and micromotion, none of which are directly captured by traditional post-mortem techniques. This study aims to characterize the localised mechanics of an ex vivo ovine osteochondral tissue–biomaterial system extracted following six weeks in vivo testing, utilising laboratory micro-computed tomography, in situ loading and digital volume correlation. Herein, the full-field displacement and strain distributions were visualised across the interface of the system components, including newly formed tissue. The results from this exploratory study suggest that implant micromotion in respect to the surrounding tissue could be visualised in 3D across multiple loading steps. The methodology provides a non-destructive means to assess device performance holistically, informing device design to improve osteochondral regeneration strategies.

## 1. Introduction

Degeneration of osteochondral tissue, and in particular articular cartilage, is increasingly prevalent and current early-stage surgical interventions do not provide satisfactory long-term outcomes to critically sized defects [1,2]. The gold standard early-stage intervention remains microfracture for which only defects smaller than 2 cm^2^ can be treated [3]. The treatment gap is particularly relevant for younger trauma patients with otherwise healthy joints. Lack of successful treatment increases susceptibility to osteoarthritis, the most common joint disease and a leading cause of pain and mobility loss worldwide [4]. The problem in early stage intervention is chiefly an inability to effectively regenerate articular cartilage [5] and the demand for osteochondral implants with tissue-regenerative capabilities has never been stronger. Devices are constantly being developed in an effort to address this currently unmet need with varying degrees of success [6,7]. The challenges and current device technologies to address regeneration have been recently reviewed for both articular cartilage [8] and the osteochondral interface [9]. Effective design requires intimate knowledge of the local tissue environment and matching of these mechanical properties with the intended device [10]. Upon implantation, device suitability and long-term success regenerating tissue relies upon suitable interaction of the tissue and implant components as a complete system. Poorly considered system design may lead to common problems such as progenitor cells not differentiating into the intended cell lineage [11,12] and failure occurring if surface geometry coupled with micromotion leads to high interfacial strain concentrations [13].

On the pathway to regulatory approval, the development and optimisation of orthopaedic devices inevitably involves the use of animal models. It is imperative to gather as much information as possible from these studies, both to test the product with maximum scrutiny, and to limit the number of animals utilised due to financial and ethical considerations. In addition to scoring systems of the defect site, retrieved explants will typically undergo destructive 2D histological analysis to determine the type and quality of tissue formed [6]. With the advent of high-resolution three-dimensional imaging techniques, such as micro-computed tomography (micro-CT), comes the potential to carry out volumetric morphological analysis as a complimentary technique to histology. Beyond passive morphometric analysis, micro-CT imaging and analysis has now progressed to quantitative mechanical testing with the pairing of in situ loading and digital volume correlation (DVC). Images captured with 3D imaging techniques such as micro-CT are combined with DVC to track features within a sample between a reference state and during mechanical loading. Features are correlated between the two or more loading states resulting in displacement vectors from which strains can be derived, non-destructively in 3D. DVC was initially developed for use with micro-CT scanning of trabecular bone 20 years ago, utilising the trabecular microarchitecture as features to create a unique pattern. The technique enabled non-destructive mechanical evaluation of the tissue and ultimately provided 3D strain maps for the first time [14]. In the last 20 years, DVC has provided the opportunity to characterise mechanics such as: strain across the osteochondral interface [15,16] examining ex vivo explants to quantify bone-implant micromotion [17], bone cement implantation [18] and cementless implant press-fitting [19]. Following sample retrieval, ex vivo analysis has taken place on bone-biomaterial systems for tissue regeneration [20], the stability of bone-screw systems [21] and newly formed bone tissue [22]. No studies thus far have considered DVC analysis of osteochondral tissue-biomaterial systems.

This study aims to investigate the potential for three-dimensional evaluation of strain and micromotion within osteochondral tissue-biomaterial systems using an implanted 3D printed hybrid biomaterial scaffold. This was carried out following six weeks of implantation within an ovine model, a typical study length and model choice for articular cartilage device research [23]. The 3D printed hybrid scaffold implant was specifically chosen as the implanted regenerative medicine device because of its ability to take cyclic load, tailorable mechanical properties which in this instance were similar to that of the articular cartilage tissue [24], the fully-degradable nature over a period of months and evidence that, when printed with pore channel sizes of 200–250 µm, human mesenchymal stem cells were sent down a chondrogenic lineage and stimulated to produce articular-like cartilage matrix in vitro [25]. Analysis was carried out using in situ micro-CT mechanical testing and DVC on an explanted tissue-biomaterial sample (n = 1) including newly formed tissue. The limited sample size makes this an exploratory study, providing a first non-destructive understanding of the mechanical conditions within an osteochondral tissue-implant system across the interface with newly formed tissue. Future studies may enable development of successful osteochondral regeneration technologies to restore joint mobility and reduce osteoarthritis prevalence. 

## 2. Materials and Methods

### 2.1. Implant Production

The scaffold implant composed of a SiO_2_-PTHF-PCL hybrid material [24] was extrusion 3D printed in a repeating structure with struts of approximately 200 µm diameter and 200 µm spacing to a height of 1.2 mm (Figure 1a). Following printing and drying, the implant was washed twice in deionised water, cut to 6 mm diameter (hollow punch, Boehm, Germany), washed again in deionised water and then sterilised by 50 kGy gamma irradiation.

### 2.2. Surgical Procedure

The in vivo study was conducted at Royal Veterinary College following institutional review board approval and was performed in accordance with UK home office regulations in line with the Animals Scientific Procedures Act (1986) and the transposed EU directive 2016/63/EU under Home Office project licence P16F4AA0A which was granted by Home Office on 5 August 2019. As part of this process, the project licence was reviewed and approved by the RVC AWERB on 10 June 2019. The skeletally mature female lowland mule sheep, of between 2 and 5 years old, was anaesthetised and a chondral defect was created in the load-bearing region of the lateral femoral condyle of the left hind stifle joint (Figure 1c) using a 5 mm diameter biopsy punch (5 mm Disposable Biopsy Punch, Kai Medical, Tokyo, Japan). Access to the lateral condyle was provided by dislocation of the patella. A 1.1 mm K-wire was drilled into the subchondral bone at the centre of the defect to release stem cells from the bone marrow and to act as a guide for reaming (Figure 1b). A reamer was utilised to liberate cartilage from the defect site and to remove a layer of subchondral bone to ensure contact with the underlying bone marrow (5 mm reamer with electric drill, Bosch, Gerlingen, Germany). A micro curette (Arthrex, Naples, FL, USA) was utilised to ensure the defect perimeter was not fibrillated. The hybrid implant was cut to size during surgery as required to fit the defect, using a combination of 5 mm diameter biopsy punch (5 mm Disposable Biopsy Punch, Kai Medical, Tokyo, Japan) and scalpel (No.11 blade, Swann-Morton, Sheffield, UK). The implant was placed into the defect and secured to the surrounding cartilage with cyanoacrylate glue (derma + flex, Chemence Medical, Alpharetta, GA, USA) at four points on the perimeter of the top of the implant. Prior to suturing, the joint was articulated through ten flexion/extension cycles to ensure the implant was secure and to provide flow of synovial fluid. The incision was sutured, and the animal transferred to a recovery area. After six weeks, the tissue-implant system of diameter approximately 15 mm diameter and approximately 8 mm height were extracted with an oscillating saw and transferred to PBS (DPBS, #14190-094, Thermo Fisher Scientific, Waltham, MA, USA) with antibiotics and antimycotics (A5955, Sigma-Aldrich, St. Louis, MO, USA) at a concentration of 1% *v*/*v*.

### 2.3. Micro-CT Imaging and In Situ Mechanics

The explanted sample was micro-CT scanned in a plastic container filled with PBS on the day of retrieval using a Versa 520 micro-CT scanner (Zeiss, Oberkochen, Germany) with coarse settings to provide a rapid evaluation of the defect site (Table 1). Each projection was of 2048 × 2048 resolution with no binning or filtering applied. Prior to in situ testing sample diameter was further reduced to approximately 10 mm using a precision sectioning saw (IsoMet, Buehler, Lake Bluff, IL, USA) to improve scan quality. The sample was stained in 1% *w*/*v* phosphotungstic acid (*w*/*v*, H_3_PW_12_O_40_, PTA, #79690 Sigma-Aldrich, St. Louis, MO, USA) solution in 70% ethanol for 21 h following established protocol [26]. The sample was mounted to the bottom polyoxymethylene platen with polymethyl methacrylate bone cement (Simplex, Kemdent, Swindon, UK). The sample and platen were loaded into a custom-built in situ mechanical testing rig equipped with a load cell (LBS100, Interface Force Measurements Ltd, Crowthorne, UK). The plastic sample chamber was filled with 70% ethanol, the top compression platen was seated and silicone grease (404-124, RS Components, Corby, UK) used around the circumference to limit evaporation of the ethanol during scanning. The bottom platen was static, and all uniaxial loading was applied to the sample by the top platen. The custom-built top platen had a recessed diameter to avoid the risk of any released or formed gas bubbles in the sample vicinity being captured during scanning (Figure 1d). A small preload (<5 N) was applied to ensure good contact with the platens and two repeat reference scans were taken consecutively and thereafter in situ mechanical testing was performed at two levels of applied compression. Prior to each scan 30 min of equilibrium time was provided with the sample within the micro-CT scanner to ensure thermal and mechanical stability. The sample was kept in liquid at all times, including during scanning (Table 1).

Reconstruction of the projection images to produce 3D volumetric data sets was performed using the Reconstructor Scout-and-Scan software (Zeiss, Oberkochen, Germany). The reconstructed CT volumes were visualized and analysed using (Fiji Is Just) ImageJ software [27] (version 1.52g, NIH, Bethesda, MD, USA). For all scans, the optional rigid body movement correction module in DaVis was utilised to provide volumetric registration. After DVC calculation, the “subtract vector at reference position” process was utilised to restore displacements to the original orientation.

To evaluate error, the repeat pair of unloaded images were considered as constant-strain conditions [28]. Within Fiji, two separate pairs of image stacks of dimensions 450 × 450 × 450 voxels were manually aligned and then extracted to include all constituent parts of the tissue-biomaterial system (Figure 2), including both the implant (Figure 2d) and local tissue (Figure 2a,e–g). Images were imported into DaVis version 8.4.0 (LaVision, Göttingen, Germany) and the constant strain study was conducted across a range of subvolume sizes from 16–112 voxels (~30–205 µm) in increments of 16 for all three DVC processing methods available in the software 1) Direct correlation (DC) 2) Fast-Fourier transform (FFT) 3) A combined FFT + DC approach. For FFT + DC, the FFT pre-shift was set at 12 pixels larger than the DC step and three calculation passes were utilised.

Error was quantified by calculating the mean absolute error (MAER, Equation (1)), representing the systematic error or accuracy, and the standard deviation of error (SDER, Equation (2)), representing the random error or precision inherent within the system [28]. MAER and SDER of strain were calculated through the cartilage using a MATLAB script (R2019a, The MathWorks Inc, Natick, MA, USA). The random error was also calculated for the displacement components. The systematic error of displacement was not calculated because the real displacement is unknown.
(1)$$MAER=\frac{1}{N}\overset{N}{\underset{k=1}{\sum }}\left(\frac{1}{6}\overset{6}{\underset{c=1}{\sum }}\left|\varepsilon_{c,k}\right|\right)$$
(2)$$SDER=\sqrt{\frac{1}{N}\overset{N}{\underset{k=1}{\sum }}{\left(\frac{1}{6}\overset{6}{\underset{c=1}{\sum }}\left|\varepsilon_{c,k}\right|-MAER\right)}^{2}}$$
where “*ε*” represents the strain; “*c*” represents each of the six independent strain components; “*k*” represents the measurement point at each subvolume and N is the number of subvolumes measured.

Following optimisation of the DVC calculation technique and subvolume size using the error quantification, the loading scenario image stacks (unloaded-load1-load2) were processed in accordance with the maximum acceptable error set as 10% of the nominal strain [15]. For this osteochondral system, the highest nominal strain applied to a component of interest was approximately 2.5%; therefore, the maximum acceptable error was set as 0.25% strain (2500 µS). Using this threshold, the loaded DVC study was carried out using the FFT + DC method and a subvolume size of 80 voxels, providing spatial vector distancing of 181 µm. Analysis under loading was carried out with a volume of interest of dimensions 821 × 821 × 401 voxels (approximately 3.7 × 3.7 × 1.8 mm) at the interface surrounding the defect.

## 3. Results

### 3.1. In Situ Mechanical Loading and Deformation Visualisation

Compressive load was successfully applied between successive scans by way of the in situ mechanical testing rig during micro-CT scanning. Analysis of micro-CT established that the subchondral bone had been penetrated during surgery, so that the implant sat under the chondral surface in a void surrounded by subchondral bone (Figure 3a). The topography of the bone created an uneven bone-cartilage interface height, which impeded compression of the cartilage (Figure 3a *). The placement of the implant, at the level of the subchondral bone, did not permit comparable compression of the implant-cartilage interface (Figure 3b,c). 

Displacement and strain were successfully calculated and visualised in three-dimensions (Figure 4). Displacement of the implant compared to the surrounding tissue was observed, giving a measure of relative micromotion (Figure 4b and e). Strain within the implant was low, of a similar magnitude to the surrounding tissue, yet high strain was measured in the defect void at both loading levels (Figure 4c,f). Of note, the correlation value was high (>0.8) for the bone, cartilage and hybrid components, yet the defect void surrounding the implant lacked sufficient pattern, hence only limited correlation was possible in these regions and thus caution must be observed in the mechanical results for these regions (Figure 4d,g).

Each slice from the three-dimensional volume provided rich localised data points of the displacement from which components of strain may be extracted as required. Total displacement and minimum principal strain were plotted (Figure 5) between the key components of the system. As observed visually, displacement was highest in the hybrid implant, with an average of 40 µm relative to the surrounding bone (Figure 5a). However, when considering strain, the hybrid implant experienced little mechanical strain compared to both the cartilage and underlying void (Figure 5b).

### 3.2. Error Quantification

Displacement and stain errors were quantified for all tissue and biomaterial components (Appendix A
Appendix A) of the system across a range of subvolume sizes and for the three DVC methods available. For the tested sample, the combined FFT + DC approach resulted in the lowest strain errors for all sample components and all subvolume sizes. The FFT approach appeared to give, on average, the lowest displacement errors. Errors at the optimal subvolume size of 80 voxels using the FFT + DC DVC method and R^2^ values for the power law equation for errors across all subvolume sizes (Figure 6). Both strain errors were below 0.08% strain for all components except for the cartilage which incurred a strain accuracy of 0.12% strain (Figure 6a,b). Displacement error for all components was below 8 µm (0.008 mm, Figure 6c). Strain error across subvolume sizes followed a power law relationship for all measured system components with R^2^ values of 0.88–0.99 (Figure 6d).

### 3.3. Micro-CT Scanning Protocol

On the day of implant retrieval, a coarse micro-CT scan of the sample was collected with the sample immersed in PBS (Figure 7a) using high power and short exposure time to rapidly collect projection data (Table 1). The subchondral bone is clearly visible (B, Figure 7a), and the hybrid implant is visualised with poor contrast (H, Figure 7a). Following coarse scanning, the sample diameter was reduced to fit within the in situ testing rig and stained with a contrast agent to increase contrast of low-density soft tissue components. A scanning protocol enabling propagation phase contrast was developed (Table 1) and used for the duration of in situ testing for DVC analysis (Figure 7b). The resulting scan (Figure 7b) enabled clear visualisation of the articular cartilage (C, Figure 7b) and in particular its boundary, in addition to tissue on and within the defect which could not been visualised with the coarse scan (Figure 7a). Note the beam hardening artefacts (Φ, Figure 7a) associated with high voltage and a high gradient in material densities which is not observed in the DVC scan protocol (Figure 7b).

## 4. Discussion

### 4.1. Key Findings

The most important finding of this study was that laboratory micro-CT and DVC enabled non-destructive mechanical testing of an osteochondral tissue-biomaterial sample retrieved following implantation within an ovine model (Figure 3, Figure 4 and Figure 5). Despite imperfect scaffold placement, the technique developed in this preliminary study provided valuable insights into relative micromotion of the system components. We present a sample preparation and micro-CT scanning protocol (Figure 7) using only laboratory equipment and avoiding the need to access synchrotrons which has been noted as a limitation [15]. Visualisation of displacement and strain within the system was possible including quantification of local strain concentrations (Figure 4) and micromotion (Figure 5). This could ultimately increase efficacy of regenerative medicine solutions by enabling optimisation of strain-matching, fixation methods to limit micromotion, and ensuring local implant behaviour is compatible with the surrounding tissue. Quantified displacement and strain errors within all tissue and biomaterial components of the system were within acceptable limits (Figure 6) [15,20]. It is possible that the magnitude of errors could be further reduced by further optimising the DVC steps [20] or image filtering [29]. 

### 4.2. Comparison to Literature

DVC has previously been carried out on bone and biomaterials tested after six weeks of implantation in sheep [20]. The study reported strain accuracy and precision of 0.02% and 0.01%, respectively, with a spatial resolution of 125 µm, yet was carried out on a bone model at a synchrotron with approximately 30% higher resolution. The same authors carried out separate studies using laboratory micro-CT and DVC to consider the properties of regenerated bone tissue [22] and of the bone-cartilage interface without an implant [16]. Our study provides a natural progression for researchers aiming to develop chondral repair and regeneration with scaffold materials. The strain accuracy and precision calculated during the constant strain error analysis followed a power law relationship with decreasing error with increasing subvolume size, as previously described [30,31]. Implant micromotion has previously been quantified using DVC on a simplified ex vivo porcine bone-implant system [17]. Our study provides a progression by considering the osteochondral interface and enabling non-destructive testing of an in vivo sample including soft tissue as part of the system.

Strain errors were approximately similar between all components of the system except for cartilage which incurred errors 2–4 times higher than the other components. Further work is required to optimise the sample preparation [26] and micro-CT scanning protocols to ensure sufficient features are imaged. Strain errors of approximately 0.1% strain or less were possible with a subvolume size of 80 voxels, providing a spatial resolution of 360 µm. When combined with 50% overlap this effectively provided a vector spacing of 180 µm. Of note, the displacement errors observed in this study were higher than would be expected. Displacement precision was calculated to be between 4 and 8 µm for the different components at the chosen subvolume size, equating to 0.9–1.8× the voxel size. The previously described sheep study observed displacement precision of 0.22 µm [20], and a laboratory micro-CT study using the same software package observed precision of less than 2 µm [18]. A recent study utilised a similar micro-CT scanning setup to measure errors in bovine bone and cartilage [16]. Using optimised settings at a subvolume size of 80 voxels strain MAER (systematic error) in cartilage was under 0.05% strain (500 µε) and in bone was under 0.04% strain (400 µε). Strain SDER (random error) was under 0.005% (50 µε) for both cartilage and bone, respectively [16]. In comparison at the same subvolume size we calculated cartilage strain MAER of 0.12% strain (1200 µε), bone MAER of under 0.04% strain (400 µε), cartilage SDER of under 0.08% (800 µε) and bone SDER of under 0.04% (400 µε). The cartilage MAER we calculated was over twice as large as in the previous study, though well within the allowable range [15]. Bone MAER was comparable between both studies whilst SDER for both components was considerable higher in our study. Scan quality, and therefore measured error, is affected by sample size and scanning voxel size. The sample tested in the presented study was approximately 10 mm diameter, three times the diameter of the samples in the previous study, and scanned a voxel size with approximately half the resolution (4.5 µm compared to the previous study at 2.02–2.56 µm); therefore, it is reasonable to expect for errors to be larger. Likewise, for the previously mentioned study of the bone-biomaterial system [20], our measured error in bone was at least twice as large yet was carried out with worse resolution on a laboratory system with a sample over twice the diameter. Further work should be considered which may further reduce the measured errors. It should be noted that the previous study avoided the use of contrast agents, which is ideal for avoiding potential changes to the system’s mechanical behaviour. Although suitable for scenarios including cartilage which typically experience high strain and displacement (approximately 10% and 150 µm), further work should be carried out to investigate why displacement precision was comparatively poor compared to strain errors.

### 4.3. Limitations

In situ mechanical testing and DVC analysis was carried out on a very limited sample size (n = 1) therefore the work is intended as an exploratory pilot study. Studies of large animal models inherently involve small sample sizes, and, likewise, DVC studies are usually limited to small sample sizes [15,22]. With optimisation the method is compatible with histological analysis which could be carried out destructively after in situ testing [26]. The developed technique ensures maximum data can be collected from each sample to inform development of successful devices.

Although standard protocol was followed including staining, the cartilage of this study did not appear to show obvious anatomical features as documented in other species [26]. This is a likely explanation of why the cartilage incurred the highest errors in the constant strain study, and further work is required to improve the cartilage imaging. To provide a control for the error and strain response of the native bone-cartilage it would have been of interest to include data for the same methodology carried out on the contralateral leg. A limitation is that this was not included as part of the presented study. Our previously documented success with cartilage imaging were carried out on smaller tissue explants (3 mm diameter) in different species to that of this trial [26]. The same protocol was followed, yet further optimisation is required to ensure successful imaging. Comparing our coarse scan with the unstained specimen compared to the longer duration propagation phase contrast scan with the stained specimen it is clear that scan quality was improved (Figure 7), yet further work is required to deduce which components of the protocol attributed to the improved quality. Staining may affect mechanical properties [32,33], previous findings suggest that it may be avoidable [26] which should be considered for future work.

The implantation site created during surgery of this pilot study resulted in removal of both cartilage and bone tissue, and therefore the implant receiving reduced mechanobiological stimuli than intended and little contact to the proposed cartilage region (Figure 3). DVC results of the explanted sample (Figure 4 and Figure 5) illustrate this: the implant, and superficial newly formed tissue, experienced displacement yet this was largely rigid body micromotion due to the underlying void rather than intended deformation. For this reason, average displacement appeared to be higher than the cartilage (Figure 5) yet resulted in very little strain compared to the surrounding tissue void. It could be inferred that limited mechanical strain was likely transduced to cells within the scaffolds during the in vivo study. The micromotion of the implant (40 µm) in comparison to the surrounding bone was within acceptable limits but in light of the implant placement further work, which this pilot study has informed, is ongoing to confirm these results within the intended chondral defect site. Micro-CT provided a rapid insight into system morphology which traditional techniques would not have provided non-destructively. This study illustrated that the described technique is able to successfully illustrate the unintended micromotion of the implant within the surrounding defect void, which itself is of great value.

## 5. Conclusions

Combining laboratory micro-CT, in situ testing and DVC provided a means to non-destructively evaluate biomechanics within the tested osteochondral tissue-biomaterials explant system. The technique provides quantification of displacement and strain, enabling measurement of load-bearing capacity between the system components and fixation to be evaluated. Despite system components not being optimally aligned for mechanical compression during this exploratory study, relative micromotion of the implant could be observed non-destructively in 3D. Engineers and materials scientists may use this technique to observe strain-matching capabilities of implants, ensuring suitable mechanobiological stimuli to cells for successful cartilage regeneration. Having the ability to non-destructively test the entire system holistically and yet quantify local micro-level deformations provides a compelling case for wide-spread application of the technique.

## Figures and Tables

**Figure 1 materials-13-03911-f001:**
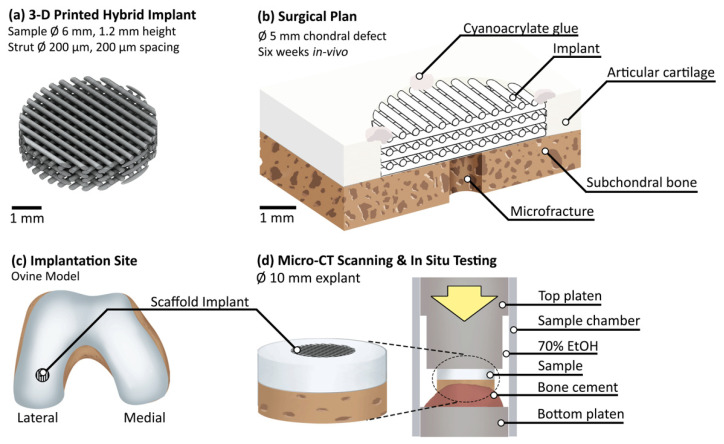
Schematics for each component of this ovine model study: (**a**) hybrid 3D printed scaffold implant design; (**b**) cross-section through surgical procedure plan, with the implant sitting within a chondral defect on top of the subchondral bone, with microfracture providing nutrient delivery and secured with cyanoacrylate glue at four points around the perimeter; (**c**) implantation site in the lateral condyle in a mature female sheep; (**d**) the extracted sample composed of the integrated hybrid implant–cartilage–bone system underwent in situ compression testing during micro-CT scanning.

**Figure 2 materials-13-03911-f002:**
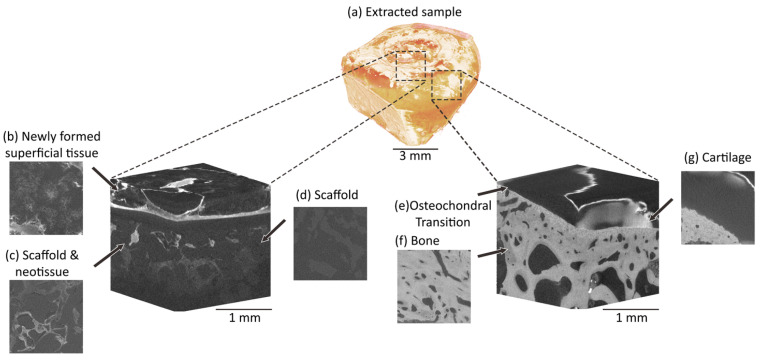
Micro-CT reconstructed volumes of the extracted sample following six weeks in vivo. The entire extracted sample (**a**) with two volumes of interest digitally extracted (indicated with dashed boxes) to encompass all system components: newly formed tissue (**b**) covered the scaffold implant (**d**), within which there were also regions of newly formed tissue (**c**). Surrounding the defect was articular cartilage (**g**), the osteochondral transition between the cartilage and bone (**e**) and the subchondral bone (**f**). These components were analysed with DVC software between pairs of repeated constant strain micro-CT scans. The constant strain study provided error analysis on all components of the tissue-biomaterial system. No image processing was been applied to the micro-CT slices.

**Figure 3 materials-13-03911-f003:**
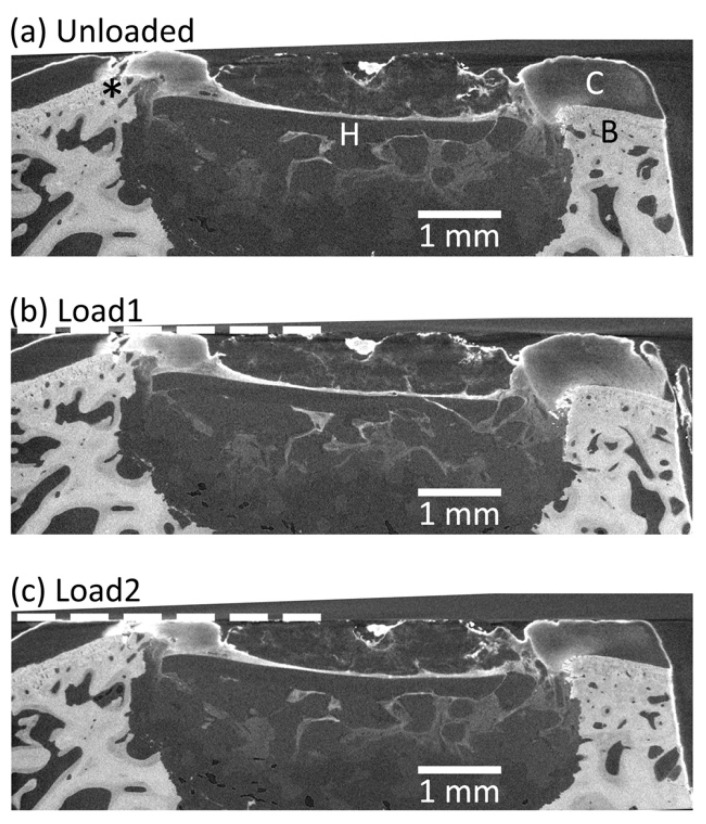
A transverse slice of the micro-CT scan of an ex vivo osteochondral sample taken from the implant site after 6 weeks, in situ tested with successive levels of load applied: (**a**) unloaded reference scan; (**b**) first level of compression; (**c**) second level of compression. The system components are labelled in panel a: C = Cartilage, B = Bone, H = Hybrid scaffold implant. A peak in the height of the bone, seen on the left-hand side of the defect (*), impeded loading of the cartilage and hybrid. The dashed line represents displacement of the compression platen.

**Figure 4 materials-13-03911-f004:**
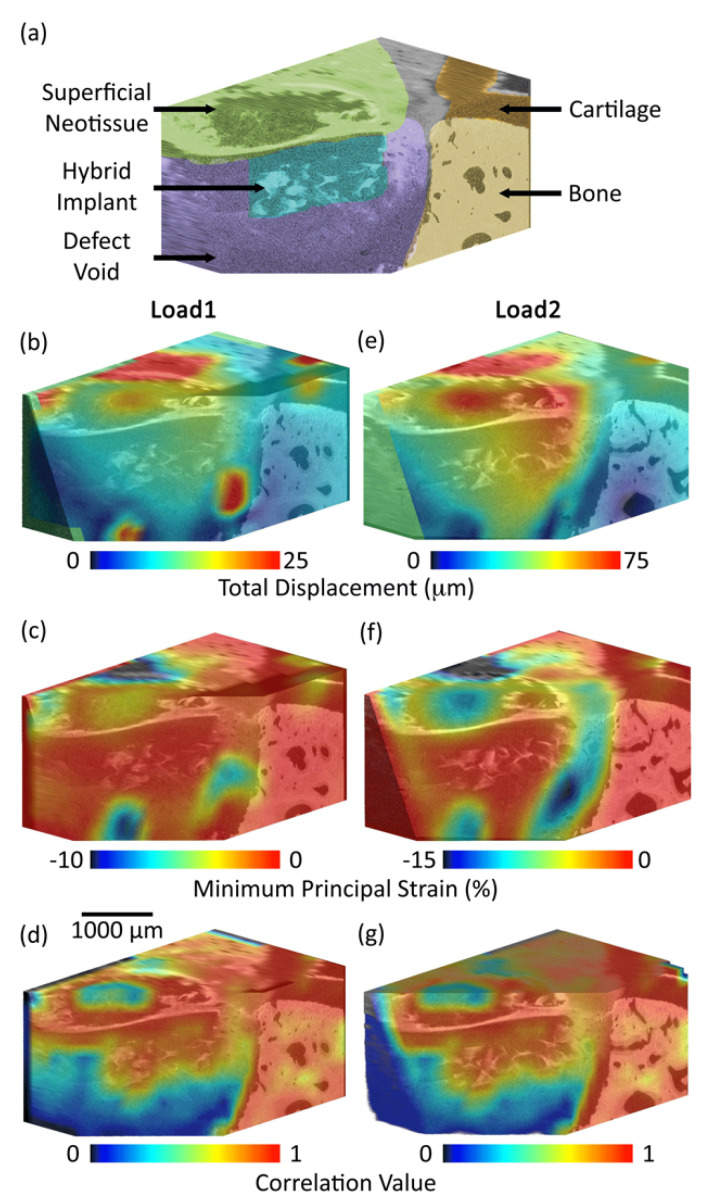
Micro-CT rendering with annotated colouring shows the system components (**a**). Combining three dimensional imaging with in situ mechanical testing and DVC enabled visualisation of local displacement (**b**,**e**) and strain (**c**,**f**) for each component of the system (**a**) at two levels of compressive loading; also presented is the correlation value across the system components (**d**,**g**): 0 = no correlation 1 = perfect correlation.

**Figure 5 materials-13-03911-f005:**
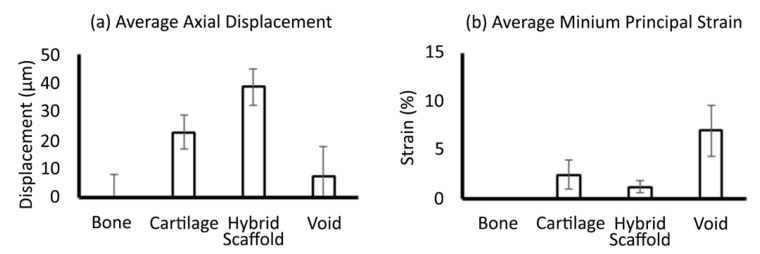
Data collected from subvolumes (n = 3) at slices (n = 5) throughout the 3D volume for total displacement (**a**) and minimum principal strain (**b**). Total displacement was normalised against bone. Average minimum principal strain in bone was less than 0.1%. Values are mean and error bars represent S.D. Data presented here represents Load2: the higher level of compression applied.

**Figure 6 materials-13-03911-f006:**
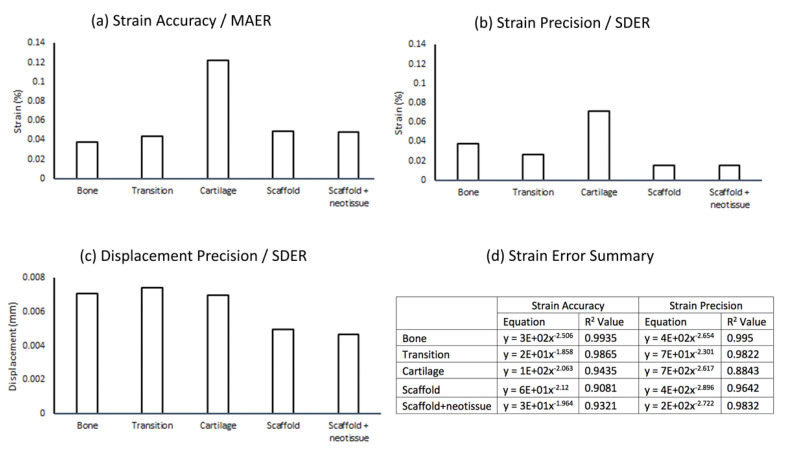
Quantification of error for five components of the tissue-biomaterial system for the evaluated sample (n = 1) at a subvolume size of 80 voxels (360 µm) using the FFT + DC method: (**a**) Strain accuracy (**b**) Strain precision (**c**) Displacement precision (**d**) Power law relationships computed for the FFT + DC method across all tested subvolume sizes for the system components where y represents the error metric and x represents the subvolume size.

**Figure 7 materials-13-03911-f007:**
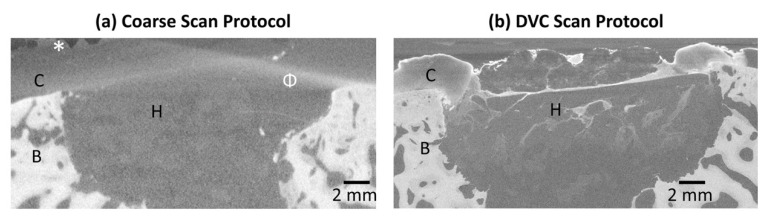
Transverse slices from two micro-CT scans of the sample following different scanning protocols: (**a**) the sample scanned in PBS on the day of retrieval with a preliminary coarse absorption contrast scan (**b**) the sample after reducing excess tissue around the perimeter, staining in 1% PTA and 70% ethanol and using a longer propagation phase contrast scanning protocol. Brightness and contrast have been approximately normalised. C = Cartilage region, B = Subchondral Bone, H = Hybrid scaffold implant. * = air bubbles relating to containment of the sample under coarse settings. Φ = beam hardening artefacts. Scale has been normalised between the two scans which were originally produced using different voxel sizes, see Table 1.

**Table 1 materials-13-03911-t001:** X-ray micro-computed tomography parameters for scanning the extracted tissue-biomaterial sample. A coarse scan was taken to provide initial evaluation of the interior interface. In situ testing was then carried out with scans taken under successive load to permit study of deformation mechanics using DVC. SOD = Source to Object Distance, ODD = Object to Detector Distance.

Scan Protocol	Voltage (kV)	Current (µA)	SOD (mm)	ODD (mm)	Voxel Size (µm)	Number of Projections	Exposure Time Per Projection (s)	Liquid Medium
**Coarse**	80	87.5	40	95	10.2	2401	5	PBS
**DVC study**	60	83	26	172	4.5	1201	16	70% EtOH

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
