# Peer review of "Exploratory Full-Field Mechanical Analysis across the Osteochondral Tissue—Biomaterial Interface in an Ovine Model"

_materials, 2020, doi:10.3390/ma13183911_

Round 1

Reviewer 1 Report

Dearest Authors,

really interesting work, worth publishing given the topic is really "hot" for those working in this field.

I would only suggest a widening of literature references, also considering some more recent works that came out, and hence a deeper discussion when comparing results with state-of-art.

best & be safe

Reviewer 2 Report

There are some significant flaws in this study and the level of innovation, nature of analysis and amount of results are not sufficient for a full paper. More detailed comments are included below.

Introduction:

- Text surrounding references [6,7] – it is suggested to cite some high quality, recent (<3 years) review papers on osteochondral scaffolds/devices to better illustrate this point.

- References [11,12] – these two references are both very old and do not reflect the current state of the literature, please replace with more recent references.

- More details on the principles and processes involved need to be given on how micro-CT imaging and DVC can be used to derive non-destructive mechanical evaluation of tissues.

- The introduction should briefly explain the study design – why was the sheep model (and only 1 animal) chosen? What is the defect representative of and what is the justification of the implant material/morphology used?

Materials and methods:

- The sentence before the start of the first sub-section is a mistake and should be removed.

- There needs to be a more detailed description of the custom-built mechanical testing equipment in the methods text on page 4, corresponding to Figure 1d. The components of the equipment shown in Figure 1d have not been explained in the text.

Results and discussion:

- The results and discussion are relatively meaningless as it is obvious that the implant did not integrate with the host tissue and sank below the chondral surface. The loading was not even conveyed to the implant and only one sample was tested. All of the ensuing analyses on mechanical parameters are therefore based on an invalid model. It is not clear if the findings of this study contribute any useful information to the current literature and innovation is very low. There needs to be at least several samples with proper placement for a valid analysis.

- There is no control for this study – why not conduct the same experiment using the contralateral leg as a comparison? Also, since the level of healing was not assessed, what is the rationale for implanting the sample for a period of 6 weeks? How might this have impacted the results?

Reviewer 3 Report

Clark et al. in their manuscript provide a novel method to quantify the displacement of implant and assess the strain within the implant that could serve as a useful tool to strain-match implants for providing an optimal mechano-biological stimulus to the chondrocytes with a goal to achieve successful cartilage regeneration. The finding that relative micromotion of an implant could be assessed three-dimensionally in a non-destructive fashion using this method in an ovine model of surgically created chondral defect is novel. The methodology, assessment techniques and analyses are performed well and the results are clearly presented. The strengths, weaknesses and comparisons to existing literature presented in the manuscript are very useful and highly recommended.

As such I find the manuscript is of importance to the field and recommend for publication.

The manuscript can be accepted for publication after revising for minor typographical and grammatical errors.

Round 2

Reviewer 2 Report

No more comments to add